# Human Fibroblast Growth Factor-Treated Adipose-Derived Stem Cells Facilitate Wound Healing and Revascularization in Rats with Streptozotocin-Induced Diabetes Mellitus

**DOI:** 10.3390/cells12081146

**Published:** 2023-04-13

**Authors:** So-Young Chang, Jun Hee Lee, Se Cheol Oh, Min Young Lee, Nam Kyu Lim

**Affiliations:** 1Beckman Laser Institute Korea, Dankook University, Cheonan 31116, Republic of Korea; so4040@hanmail.net; 2Institute of Tissue Regeneration Engineering (ITREN), Dankook University, Cheonan 31116, Republic of Korea; junheelee@dankook.ac.kr; 3Stem Cell R&D Center, N-BIOTEK Inc., Bucheon 14449, Republic of Korea; sco61111@naver.com; 4Department of Otolaryngology-Head & Neck Surgery, College of Medicine, Dankook University, Cheonan 31116, Republic of Korea; 5Department of Plastic and Reconstructive surgery, College of medicine, Dankook University, Cheonan 31116, Republic of Korea

**Keywords:** diabetes mellitus, diabetic foot disease, wound healing, adipose-derived stem cells

## Abstract

Diabetes mellitus contributes to 15–25% of all chronic foot ulcers. Peripheral vascular disease is a cause of ischemic ulcers and exacerbates diabetic foot disease. Cell-based therapies are viable options to restore damaged vessels and induce the formation of new vessels. Adipose-derived stem cells (ADSCs) have the potential for angiogenesis and regeneration because of their greater paracrine effect. Preclinical studies are currently using other forced enhancement techniques (e.g., genetic modification or biomaterials) to increase the efficacy of human ADSC (hADSC) autotransplantation. Unlike genetic modifications and biomaterials, many growth factors have been approved by the equivalent regulatory authorities. This study confirmed the effect of enhanced human ADSC (ehADSC)s with a cocktail of FGF and other pharmacological agents to promote wound healing in diabetic foot disease. In vitro, ehADSCs exhibited a long and slender spindle-shaped morphology and showed significantly increased proliferation. In addition, it was shown that ehADSCs have more functionalities in oxidative stress toleration, stem cell stemness, and mobility. In vivo, the local transplantation of 1.2 × 10^6^ hADSCs or ehADSCs was performed in animals with diabetes induced by STZ. The ehADSC group showed a statistically decreased wound size and increased blood flow compared with the hADSC group and the sham group. Human Nucleus Antigen (HNA) positive cells were observed in some ADSC-transplanted animals. The ehADSC group showed a relatively higher portion of HNA-positive animals than the hADSC group. The blood glucose levels showed no significant difference among the groups. In conclusion, the ehADSCs showed a better performance in vitro, compared with conventional hADSCs. Additionally, a topical injection of ehADSCs into diabetic wounds enhanced wound healing and blood flow, while improving histological markers suggesting revascularization.

## 1. Introduction

Diabetes mellitus contributes to 15–25% of all chronic foot ulcers [1]. Diabetic patients with neuropathy and peripheral vascular disease typically do not recognize the trauma. Peripheral vascular disease is a cause of ischemic ulcers and exacerbates diabetic foot disease. Peripheral vascular disease (alone or in combination with diabetes mellitus) can culminate in recurrent, nonhealing ulcers and amputations. Approximately 50% of patients with diabetic foot disease have concurrent vascular disease [2,3]. Because surgical revascularization is not always feasible in these patients, there is an urgent need for alternative therapies that can improve the blood supply to the ischemic foot. However, such improvement is difficult in diabetic patients who have problems with re-epithelialization and wound healing related to reduced blood circulation and decreased fibroblast migration [4,5]. Cell-based therapies are viable options to restore damaged vessels and induce the formation of new vessels.

Stem cells promote wound healing through anti-inflammatory effects, angiogenesis, growth factor release and differentiation into the various cells that are involved in wound healing [6,7]. Stem cell therapy using mesenchymal stem cells has been investigated in both preclinical and clinical studies [8,9]. The autologous application of stem cells has multiple advantages [10]. Furthermore, stem cells can be isolated from tissues in various locations throughout the body [11,12].

Compared with bone marrow-derived stem cells, adipose-derived stem cells (ADSCs) have greater potential for angiogenesis and regeneration because of their greater paracrine effect [13,14,15,16]. Clinical studies using adipose-derived stem cells are slow, and preclinical research is hindered by methodological difficulties. A few preclinical studies using mouse ADSCs demonstrated favorable results [17]. Preclinical studies are currently using other forced enhancement techniques (e.g., genetic modification or biomaterials) to increase the efficacy of ADSC autotransplantation [15,18,19,20,21]. Recently, results of type 2 diabetes studies using ADSC have also been reported [22,23].

Growth factors enhance cell signaling, proliferation, and differentiation [24,25,26]. Unlike genetic modifications and biomaterials, many growth factors have been approved by the United States Food and Drug Administration and the equivalent regulatory authorities of other countries [27]. Fibroblast growth factor (FGF) regulates multiple biological processes, and some of its subtypes are approved for clinical application [28].

The effective therapeutic option for diabetic wounds is not clear yet, and stem cell application is one of the most promising treatments to overcome the limitations. It has been reported that FGF stimulates wound healing in mice with genetic diabetes [29]. FGFs have been known to enhance the pluripotency of stem cells. In addition, FGF can promote ADSC endothelial differentiation [30]. Therefore, we assumed that cells enriched by ADSCs with FGF would have a more positive effect on wounds, especially in DM animals.

Therefore, in this study, we used ADSCs to promote wound recovery in diabetic foot disease. We used a cocktail of human recombinant FGF and other pharmacological agents to enhance human ADSCs. The enhanced hADSCs (ehADSCs) showed a better performance in vitro, compared with conventional hADSCs. Additionally, a topical injection of ehADSCs into diabetic wounds enhanced wound healing and blood flow, while improving histological markers; these findings suggested revascularization.

## 2. Materials and Methods

### 2.1. Tissue Harvest

A 50-year-old female patient who provided informed consent prior to surgery participated in this study. Adipose tissue (20 g) was harvested from the patient’s abdomen during deep inferior epigastric perforator free-flap surgery. According to ‘The Guideline on Eligibility Determination for Donors of Cell Therapy Product in 2021′ from the Ministry of Food and Drug Safety, hematological infections should be excluded. She had no hematological infection such as hepatitis B or C, syphilis, or human immunodeficiency virus. To maintain viability, adipose tissue mechanically collected using scissors was wrapped in a saline-soaked gauze, sealed, and stored in a low-temperature ice bag; it was subsequently carried from the operating room to the N-Biotek (Bucheon, Republic of Korea) laboratory.

### 2.2. Cell Isolation and Primary Culture of ehADSCs

The isolation and culture of ehADSCs were isolated by receiving adipose tissue collected from Dankook University Hospital. The adipose tissue was washed three times with phosphate-buffered saline (PB + S; Lonza, Basel, Switzerland) and digested with 0.1% (*w*/*v*) type I collagenase (Nordmark, Uetersen, Germany) for 45 min at 37 °C [31,32]. Undigested tissue and unnecessary oil were filtered out using a screen mesh, followed by centrifugation at 1500 rpm for 5 min. The resulting pellet was resuspended in complete culture medium consisting of α-Minimum Essential Medium (Welgene, Gyeongsan, Republic of Korea) that contained 9% fetal bovine serum (Thermo Fisher Scientific, Waltham, MA, USA) and 1 ng/mL bFGF (R&D Systems, Minneapolis, MN, USA). hADSCs (without enhancement) were resuspended and cultured in α-Minimum Essential Medium (Welgene) with 9% fetal bovine serum (Thermo Fisher Scientific). The cells were plated at a density of 5 × 10^3^/cm^2^ in fresh complete medium at 37 °C in a humidified atmosphere with 5% CO_2_. Twenty-four hours after isolation, unattached cells were removed by washing, and the medium was changed every 3 days (Figure 1). The characterization of ADSCs were assessed by flow cytometry (BD, Franklin Lakes, NJ, USA) on ADSC markers including CD29, CD73, and CD105 (Thermo Fisher Scientific) (Appendix A). The possibility of mycoplasma contamination was screened by a LookOut Mycoplasma PCR detection Kit (Sigma-Aldrich, St. Louis, MO, USA) in accordance with the manufacturer’s protocol (Appendix A).

### 2.3. ehADSC Expansion

ehADSC samples from primary culture were subcultured with trypsin/ethylenediaminetetraacetic acid (Sigma-Aldrich), then cultured in complete medium at 37 °C in a humidified atmosphere with 5% CO_2_. Cells were passaged upon reaching 90% confluence. Passage 2 cells were used in subsequent experiments [33].

### 2.4. Characterization of ehADSCs

Passage 2 ehADSCs were characterized by flow cytometry to examine the presence of mesenchymal stem cell-surface markers. ehADSCs were stained with antibodies for CD14-PE, CD19-FITC, CD34-FITC, CD45-APC, Anti-HLA-DR-APC A750, CD73-PE, CD90-FITC, and CD105-Cy5.5 (BD, Franklin Lakes, NJ, USA). The cells were incubated with the corresponding fluorochrome-conjugated antibodies at room temperature for 10 min. The cells were washed with wash buffer and pelleted by centrifugation. The pellets were resuspended and analyzed by flow cytometry (Beckman Coulter, Brea, CA, USA). Data acquisition and analysis were performed using the CytExpert software (Beckman Coulter) [34].

### 2.5. Endotoxin Assay

Endotoxin levels were assayed using the Kinetic-QCL Chromogenic LAL Assay Kit (Lonza), in accordance with the manufacturer’s instructions. Briefly, 200 μL of sample were suspended in 800 μL of LAL water and agitated in a vortex mixer for 60 s. Endotoxin standard solution was tenfold serially diluted (5.0, 0.5, 0.05, and 0.005 EU/mL). Next, 100 μL of sample, endotoxin standards, and blank were added in duplicate to a 96-well microplate. After the microplate had been prewarmed at 37 °C for 10 min, 100 μL of lysate solution was added to each well. During the reaction of LAL reagents, absorbance (405 nm) was read using a microplate reader and the WinKQCL Software (Lonza). We constructed a log/log linear correlation of the reaction time of each standard with the corresponding endotoxin concentration. The endotoxin concentrations of test samples were automatically calculated using a standard curve [35].

### 2.6. Sterility Test

The samples were aseptically spread over the surface of tryptone soy agar (Young Science INC., Bucheon, Republic of Korea) and Sabouraud Chloram agar (Young Science INC.). The samples were incubated at 30–35 °C on tryptone soy agar and 20–25 °C on Sabouraud Chloram agar for ≥5 days.

### 2.7. Mycoplasma Detection Assayt

The ehADSCs were confirmed to be mycoplasma-free using a MycoAlert Mycoplasm Detection Assay Kit (Lonza). The test procedure recommended by the manufacturer’s instructions was as follows: 100 μL of the sample, positive control, and negative control were prepared in each test tube. The MycoAlert Reagent was added to each test tube and reacted at room temperature for 5 min. After the reaction for 5 min, Reading A was measured on each test tube using a luminometer (Berthold, Germany). The mycoAlert Substrate was added to each test tube and reacted for 10 min at room temperature. Then, Reading B was measured on each test tube using a luminometer. The value of B/A was calculated with the Reading A and Reading B records [36].

### 2.8. In Vitro Study

#### 2.8.1. BrdU Assay

After culturing of hADSCs and ehADSC (5000 cells/well) on 96-well plate (SPL Life Sciences, Pocheon, Republic of Korea) for 24 h, a BrdU assay was performed to confirm the cell proliferation using a BrdU Cell Proliferation Assay kit (Cell Signaling Technology, Danvers, MA, USA) in accordance with the manufacturer’s protocol.

#### 2.8.2. Cell Viability Assay

After culturing of hADSCs and ehADSC (5000 cells/well) on 96-well plate (SPL Life Sciences) for 24, cell viability was examined using a Cell Counting Kit-8 (CCK-8; Dojindo, Kumamoto, Japan) after treatment with hydrogen peroxide (0, 200, and 300 μM) for 24 h, in accordance with the manufacturer’s instructions.

#### 2.8.3. Western Blot Analysis

Total cellular proteins of hADSCs and ehADSCs were extracted using radioimmunoprecipitation assay lysis buffer (Thermo Fisher Scientific). Cell lysates were subjected to sodium dodecyl sulfate-polyacrylamide gel electrophoresis, and proteins were transferred to nitrocellulose membranes (Sigma-Aldrich). Membranes were blocked with 5% bovine serum albumin (Sigma-Aldrich), then incubated with primary antibodies against Oct4 and β-actin (Cell Signaling Technology). After the membranes had been incubated with peroxidase-conjugated secondary antibodies (Cell Signaling Technology), bands were visualized using enhanced chemiluminescence reagents (Thermo Fisher Scientific) and an iBright CL1500 imaging system (Thermo Fisher Scientific).

#### 2.8.4. Migration Assay

To assess the migration of hADSCs and ehADSCs, a wound scratch assay was performed. Cells were seeded onto six-well plates (SPL) and grown until confluence. To suppress the proliferation of hADSCs and ehADSCs in wound healing scratch assay, confluent cells were treated with mitomycin C (10 μg/mL; Sigma-Aldrich) for 2 h [37]. Cell monolayers were wounded using a cell scraper, and detached cells were removed by washing with medium. After 10 h of incubation, migrated cells were observed under a light microscope (Olympus, Tokyo, Japan).

### 2.9. In Vivo Study

#### 2.9.1. Animals and Streptozotocin (STZ)-Induced Diabetes

Twenty 6-week-old adult male Sprague–Dawley rats (Narabiotec Co., Ltd., Seoul, Republic of Korea) were used for the in vivo experiments. The rats were divided into a sham group (*n* = 10), hADSC group (*n* = 7), and an ehADSCs group (*n* = 10). The sham group is a group in which 0.9% normal saline was injected into the wound in the same amount as the hADSC and ehADSC injection.

The rats were completely anesthetized with alfaxalone (Jurox Inc., North Kansas City, MO, USA) and xylazine (Bayer, Leverkusen, Germany), mixed at a 1:3 ratio (0.15 mL/100 g) and intramuscularly injected into the femoral region. All animal experiments were performed in accordance with the protocol approved by the Animal Care and Use Committee of Dankook University (DKU-22-069) and Dankook University Hospital, Republic of Korea. Institutional Review Board Guide-lines (DKUH-2020-05-001).

Diabetes was induced in rats by a single intraperitoneal injection of STZ (65 mg/kg; Sigma-Aldrich). The concentration of STZ used at this time refers to a previously published paper on how to induce DM in rats with a single dose [38,39]. STZ solution in 0.1 M citrate buffer (pH 4.5) was injected within 10 min after preparation. Diabetes induction was confirmed by measurements of blood glucose levels at 2 weeks after STZ injection. The mean blood glucose levels were 368 mg/dL in the sham group and 363.5 mg/dL in the experimental group. Diabetes induction was assessed as described elsewhere [38,39].

In order to amputate the animal’s legs, intramuscularly inject a mixed anesthetic solution (0.15 mL/100 g) of alfaxalone (Jurox Inc.) and xylazine (Bayer) in a ratio of 1:3 to completely anesthetize the animal and then cut the leg at the wound site with cutting scissors (PEACE Korea, Incheon, Republic of Korea). After sampling, the animals were sacrificed with CO_2_ gas and then carcassed.

#### 2.9.2. Wound Healing Model and ehADSC Local Injection

Wounding and ehADSC transplantation are illustrated in Figure 2. After the induction of diabetes, tacrolimus (FK-506, 1 mg/kg, Sigma-Aldrich), and dexamethasone (1 mg/kg, Sigma-Aldrich) were administered six times to minimize the risk of rejection. Wound surgery and the local injection of ehADSCs were performed on the fourth day of immunosuppressant administration. A circular full thickness wound with a diameter of 6 mm was constructed on the dorsum of the rat’s left foot using a biopsy punch (Kai Industry Co., Ltd., Seki, Japan) after the rats had been completely anesthetized.

Next, 1.2 × 10^6^ ehADSCs were injected, half each in the vascular pathway area and near the wound site (Figure 2B). The quantity of cells for injection was based on several previously reported publications because the therapeutic range of cell therapy has not yet been defined. There was a wide range of cell counts reported in the literature, from 1.0 × 10^6^ to 2.4 × 10^6^. Furthermore, the total number of cells 1.2 × 10^6^ was estimated to be more than 200 million cells in a human clinical trial which was predicted to offer sufficient therapeutic improvement [40,41,42].

#### 2.9.3. Wound Closure Measurement Blood Flow Evaluation

Wound sizes were estimated using Vernier calipers (Mitutoyo, Kawasaki, Japan) at 4 and 8 days after surgery and ehADSC transplantation. The wound area was calculated as width × height × 3.14 mm^2^ [43,44]. The blood flow in wounds was evaluated by a laser Doppler (Perfusion Imager System PeriScan PIM 3 System; Perimed AB, Stockholm, Sweden) on days 4 and 8 after transplantation. After a rat had been anesthetized, its wounded foot was placed on a black pad and scanned. Mean perfusion in the wound area was automatically calculated based on the color histogram of the laser Doppler color image [45].

#### 2.9.4. Histological Analysis

On days 4 and 8 after the local injection of hADSCs and ehADSCs, the cell-injected left feet of completely anesthetized rats were amputated and fixed in 4% paraformaldehyde (Biosesang, Seongnam, Republic of Korea) overnight. Next, decalcification was performed for 5 weeks in 0.5 M ethylenediaminetetraacetic acid (pH 8) (GeneAll Biotechnology Co., Ltd., Seoul, Republic of Korea) at room temperature. The samples were washed with 1 × PBS (Biosesang) for 4 h to terminate decalcification; crystallization was prevented by soaking the tissue in 10%, 20%, and 30% (*w*/*v*) sucrose (Sigma-Aldrich). Next, the samples were transferred to cryomolds and embedded in an OCT compound (Tissue-Tek, Torrance, CA, USA). Six-micron-thick sections were cut using a cryomicrotome (Leica; Wetzlar, Germany), then stained with hematoxylin and eosin (H&E) and Masson’s trichrome (BBC Biochemical™, Mount Vernon, WA, USA). The stained sections were observed under a microscope (BX53; Olympus) to evaluate histological changes.

#### 2.9.5. Epifluorescence Analysis

Frozen section slides were washed three times with 1× PBS for 10 min, then blocked for 1 h at room temperature with 5% normal goat serum (Vector Laboratories, Burlington, CA, USA) and 0.3% Triton X-100 (Sigma-Aldrich) to prevent nonspecific binding. For the identification of endothelial cells inside the wound, the slides were incubated with anti-endothelial cell antibody (RECA-1; Abcam, Cambridge, UK) overnight at 4 °C. Next, the slides were washed three times with PBS for 5 min, then incubated with a secondary antibody (Alexa Fluor 568-conjugated goat anti-mouse IgG; Thermo Fisher Scientific) for 1 h. The nuclei were stained using 4′,6-diamidino-2-phenylindole (DAPI; Sigma-Aldrich). Epifluorescence using anti-human nuclear antigen (HNA; abcam) was performed to confirm the presence of live ehADSC at the wound site, and 594-conjugated goat anti mouse IgG1 kappa (abcam) was used as a secondary antibody.

A TUNEL assay (BioActs, Incheon, Republic of Korea) was performed in accordance with the manufacturer’s protocol. Briefly, the slides were washed three times for 10 min in 1× PBS, then permeabilized by incubation for 5 min at 25 °C in 0.2% Triton X-100 in PBS. The slides were washed three times for 5 min with 1× PBS, then incubated with 100 μL of 1× reaction buffer for 10 min at 25 °C. PBS was removed, staining reagent (50 μL per 5 cm) was added, and slides were incubated in a humidified chamber at 37 °C for 1 h in the dark. Stained slides were immersed in 2× rinse buffer for 15 min at 25 °C, then washed with PBS for 5 min; the procedure was repeated twice to remove the unreacted dye-dUTP. Finally, the slides were stained with propidium iodide (1 μg/mL) at 25 °C in the dark for 15 min and washed three times. Images were obtained using a confocal microscope (Olympus) after the slides had been mounted in VECTASHIELD medium (Vector Laboratories).

### 2.10. Statistical Analysis

In vitro data are expressed as means ± standard errors of the mean. The results were analyzed by a one-way analysis of variance for multiple comparisons or Student’s *t*-test for paired comparisons. In vivo results are expressed as means ± standard deviations and were analyzed using Prism software (GraphPad, La Jolla, CA, USA) or SPSS software (IBM, Armonk, NY, USA). Shapiro–Wilk tests were performed to determine whether the data exhibited normality (i.e., whether parametric or nonparametric analyses should be used). Two-way repeated-measure analyses of variance (ANOVAs) with Bonferroni multiple comparisons tests were used to compare serial body weights and blood glucose levels. Two-tailed Mann–Whitney U tests (nonparametric) and unpaired *t*-tests (parametric; with/without Welch’s correction) were performed to compare blood flow, wound size, histological markers, and epifluorescence parameters among the ehADSC, hADSC, and sham groups. A value of *p* < 0.05 was considered indicative of statistical significance.

## 3. Results

### 3.1. Enhanced Characteristics of ehADSCs In Vitro

Previously, our studies reveal that senescent MSCs show the enlarged cell size [46,47]. To assess the cell size alteration depending on culture conditions, the morphological analysis was investigated. ehADSCs exhibited a long and slender spindle-shaped morphology, whereas hADSCs showed the enlarged and flattened cell size. ehADSCs exhibited a long and slender spindle-shaped morphology (Figure 2A). The proliferation of ehADSCs was significantly greater (two-tailed unpaired *t*-test, t = 11.6089, df = 22, *p* < 0.0001) than the proliferation of hADSCs (Figure 2B). hADSC viability decreased in an H_2_O_2_ concentration-dependent manner (one-way ANOVA: 0 μM vs. 200 μM, *p* = 0.0083; 0 μM vs. 300 μM, *p* < 0.0001), whereas ehADSC viability was maintained (one-way ANOVA: 0 μM vs. 200 μM, *p* = 0.9964; 0 μM vs. 300 μM, *p* = 0.2801) (Figure 2C). Western blotting indicated that Oct4 expression was significantly greater in ehADSCs than in hADSCs (two-tailed unpaired *t*-test, t = 5.453, df = 4, *p* = 0.0055) (Figure 2D). Additionally, ehADSCs showed significantly greater mobility, compared with hADSCs (two-tailed unpaired *t*-test, t = 10.02, df = 16, *p* < 0.0001) (Figure 2E). These results suggested that ehADSCs have greater efficacy than hADSCs. These enhanced effects were maintained until the 7th passage of cells (Appendix A).

### 3.2. Survival and Blood Glucose Level

After local injections of ehADSCs and hADSCs or normal saline (sham group) in the foot near the artificial wound (Figure 3A,B), body weight changes were similar at all time points (two-way ANOVA, column factor *p* = 0.4700, F(6, 45) = 0.9492; Bonferroni’s multiple comparisons test; all *p*-values not significant) (Figure 3B). For 25 days, blood glucose levels did not significantly differ among the two groups. At 29 days, the blood glucose level is almost same between ehADSC and hADSC. The ehADSC and hADSC groups (~400 mg/dL) showed lower blood glucose than in the sham group (>600 mg/dL). Despite the difference, it was not statistically significant. (Figure 3C). There was no skin tumor formation near the site of ehADSC and hADSC injection. All hADSC and ehADSC-injected animals survived for 8 days.

### 3.3. ehADSCs Enhance Wound Closure and Blood Flow

At 4 days after injection, a significant difference in the wound size was observed between the hADSC and ehADSC groups and blood flow showed the highest value in the ehADSC group, but it was not statistically significant. On day 4, the mean wound size in the ehADSC group was 121.1 ± 22.9 mm^2^, the hADSC group was 181.9 ± 15.1 mm^2^, and the sham group was 146.3 ± 49.4 mm^2^, (one-way with Tukey’s multiple comparisons test: sham vs. hADSC, mean diff = −8.916, *p* = 0.3244; sham vs. ehADSC, mean diff = −6.285, *p* = 0.2299; hADSC vs. ehADSC, mean diff = 15.20, *p* = 0.0489). On day 8, the mean wound size in the ehADSC group was 85.8 ± 29.5 mm^2^, the hADSC group was 133.6 ± 32.9 mm^2^, and the sham group was 130.0 ± 14.3 mm^2^, (one-way with Tukey’s multiple comparisons test: sham vs. hADSC, mean diff = −0.8832, *p* = 0.9737; sham vs. ehADSC, mean diff = 11.07, *p* = 0.0127; hADSC vs. ehADSC, mean diff = 11.95, *p* = 0.0261) (Figure 4B). The ehADSC group had significantly greater blood flow in the wound area, compared with the sham group and hADSC on day 8 (sham vs. ehADSC, *p* = 0.0389; hADSC vs. ehADSC, *p* = 0.0286) (Figure 4C).

### 3.4. Soft Tissue Thickness and Collagen Deposition

Soft tissue refers to the tissue that connects and supports other tissues and surrounds the body’s organs, including tendons, muscles, skin, fat, and fascia. In H&E- and Masson’s trichrome-stained sections, a tightly connected tissue structure and high collagen deposition were observed in the hADSC group, compared with the sham group, on day 8 (Figure 5A). In H&E-stained images, the distance between the middle bone and epithelial lining in the ehADSC group was 2.75 ± 0.3 mm, whereas it was 3.43 ± 0.3 mm in the sham group (two-tailed unpaired *t*-test with Welch’s correction: sham vs. ehADSC, t = 2.441, df = 9, *p* = 0.0373) (Figure 5B).

### 3.5. ehADSCs Increase the Number of Endothelial Cells and Reduce TUNEL-Positive

Immunofluorescence staining showed that RECA-1 staining intensity was higher after the ehADSC injection in diabetic wounds (6.91 ± 1.0/μm^2^) than in the sham group (3.12 ± 0.5/μm^2^) on day 8 (Figure 6A). RECA-1 staining intensity in the wound area was significantly greater in the ehADSC group than in the sham group (two-tailed Mann–Whitney U test: sham vs. ehADSC group, Mann–Whitney U = 0, *p* = 0.0286) (Figure 6C). The TUNEL assay revealed fewer TUNEL-positive cells in the ehADSC group than in the sham group on day 8 (Figure 6B). In superficial skin and deep soft tissue, the number of TUNEL-positive cells was 11 ± 3.8/10^4^ μm^2^ in the ehADSC group, whereas it was 27 ± 9.7/10^4^ μm^2^ in the sham group (two-tailed unpaired *t*-test with Welch’s correction: sham group vs. ehADSCs, t = 3.239, df = 5.221, *p* = 0.0216) (Figure 6D). Increased RECA-1 expression was consistent with the Doppler measurements of blood flow, suggesting enhanced vascularization.

### 3.6. Observation of HNA-Postive Cells

Epifluorescence analysis using HNA was performed. In both the hADSCs and ehADSCs group, some animals showed HNA-positive cells which are shown red in Figure 7. On day 4, the hADSC group showed the presence of HNA-positive cells in one animal among a total of three and the ehADSC group showed the presence of HNA-positive cells in two animals among a total of three. On day 8, hADSC showed the presence of HNA-positive cells in one animal among a total of four and the ehADSC group showed the presence of HNA-positive cells in two animals among a total of three. These results indicate that relatively more animals showed HNA-positive cells in the ehADSC group (Table 1).

## 4. Discussion

Vascular pathology-related foot disease in diabetic patients is difficult to manage because of reduced blood circulation and fibroblast migration. The treatment of foot disease in diabetic patients requires the restoration of damaged blood vessels, which may be facilitated by cell-based therapies. Our findings indicate that ehADSCs can promote the healing of diabetic wounds.

Mesenchymal stem cells can be used to repair tissue damage, and ADSCs have potential uses in regenerative medicine and wound repair [15,48]. ADSCs increase the expression levels of angiogenic factors, vascular endothelial growth factor, and bFGF in various conditions [30,49]. These growth factors are important for wound healing. In the present study, ehADSCs showed significantly greater proliferation and migration, compared with hADSCs. Liu et al. reported bFGF secretion by hADSCs under various culture conditions in vitro [49]. ADSCs inhibit oxidative stress [34], and the ehADSCs used in this study differentiated into cells with enhanced antioxidant activity.

To determine their effects on wound healing in diabetic patients, ehADSCs were locally injected into STZ-induced diabetic wounds. The ehADSC group showed faster tissue regeneration, compared with the sham group. Additionally, the blood glucose level was significantly lower in the ehADSC group than in the sham group. Blood glucose control was achieved in diabetic mice using animal tissue-derived ADSCs [50]. In another wound healing study, human-derived ADSCs promoted tissue regeneration but did not affect the blood glucose level in diabetic mice [51]. In contrast, our ehADSCs modulated the blood glucose level, suggesting its potential for use in diabetes treatment.

Wound healing requires the re-epithelialization of tissues and their functional restoration, a process that involves collagen and growth factors (e.g., vascular endothelial growth factor and bFGF). The presence of ehADSCs increased collagen synthesis around the wound, compared with the sham group. Moreover, blood flow around the wound was greater in the ehADSC group; the expression of RECA-1, an angiogenic factor, was significantly increased compared with the sham group. Therefore, angiogenesis promotes tissue regeneration during wound healing.

A typical phenomenon of apoptosis is DNA fragmentation. By labeling the area, cells undergoing apoptosis can be identified. The TUNEL assay performed in this study was most commonly used to detect cells undergoing apoptosis, a form of programmed cell death. Our results might suggest that ehADSCs inhibited apoptosis in tissues around wounds; they restored tissues via blood vessel regeneration. Still, further analysis of molecular markers for apoptosis should be performed in the near future. Although the mechanism underlying the therapeutic effect of ehADSCs warrants further investigation, Kue et al. reported that ADSCs exert autocrine and paracrine effects to heal diabetic wounds [16].

Previous studies of wound healing in animals have used mouse-derived ADSC cells or genetically modified cells; these cells exerted beneficial effects on wound healing.

Stem cell therapy, including ADSC, is recognized as an approach for tissue repair, including wound healing. In particular, human ADSCs have advantages over other types of stem cells, such as cell proliferation capacity, a toleration to oxidative stress resulting in efficacy in wound healing. In this study, we compared ehADSC (functionally enhanced with growth factors) and hADSC. Although the effect of hADSC alone was already proven, the results of the current study showed that the effect was almost the same and was not statistically significant in relation to the control (sham). This could be due to the difference of cell preparation for the in vivo transplantation process. A previous study used a commercial cell line obtained from a company (Invitrogen) for the transplantation of ADSC, but in the present study we used adipose tissue obtained from a doner. They injected 2.5 × 10^6^ cells into mice, but we used rats (~10-fold greater body weight) and injected 6.0 × 10^5^ cells. We used ~67% fewer cells than Seo et al. [51].

In the present study, we enhanced the efficacy of human ADSCs by modifying a medium composition and demonstrated statistically significant differences both in vitro and in vivo. There are limitations in this study. Since the therapeutic range of cell therapy products has not yet been defined, it is not accurate to select the dose of ehADSC to be injected. It is not clear whether the animal model used in the present study is the best option. It is possible that a genetic mouse model with better DM modeling and less invasiveness (the current STZ injection could be quite invasive) would be better. In addition, since we are trying human-derived cells on animal models, it is not easy to analyze the translational studies targeting humans. In addition, this study did not focus on the mechanical properties in a context of migration, but we have focused on the migration of ADSCs depending on different culturing methods. In the further study, the mechanical properties of a migration test in hADSCs and ehADSCs need to be investigated. Another limitation of this study was the large error bar in the experimental results”.

## 5. Conclusions

FGF-induced ehADSCs showed better functional efficacy in vitro compared to hADSCs. Moreover, a topical injection of ehADSC into diabetic wounds promotes wound healing and blood flow, possibly inhibits tissue apoptosis, and suggests revascularization.

## Figures and Tables

**Figure 1 cells-12-01146-f001:**
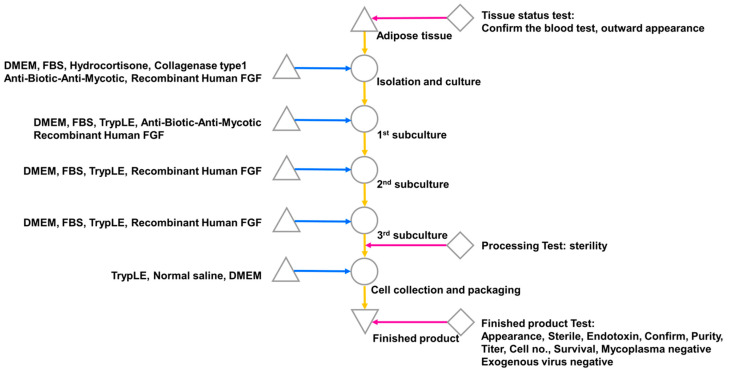
ADSC isolation from human-derived adipose tissue. Single cells were isolated from adipose tissue and subcultured three times with FGF; cells were collected after the third subculture and subjected to a product suitability test.

**Figure 2 cells-12-01146-f002:**
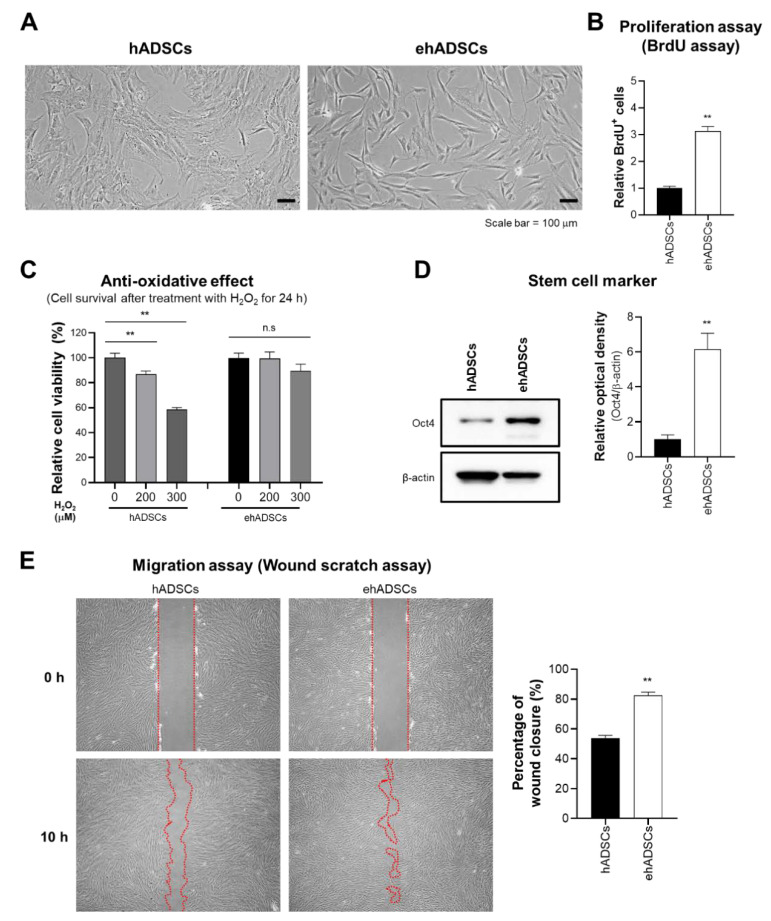
Characteristics and functionalities of enhanced hADSCs (ehADSCs). (**A**) Morphologies of hADSCs and ehADSCs. (**B**) ehADSCs showed greater proliferation, compared with hADSCs. (**C**) ehADSCs tolerated oxidative stress better than hADSCs. (**D**) ehADSCs showed more stem cell characteristics, compared with hADSCs. (**E**) Red mark area indicates the area of scratched wound site. Migration of ehADSCs and hADSCs. ehADSC mobility was significantly greater, compared with hADSC mobility. Values are means ± standard errors of the mean (** *p* < 0.01; n.s, not significant).

**Figure 3 cells-12-01146-f003:**
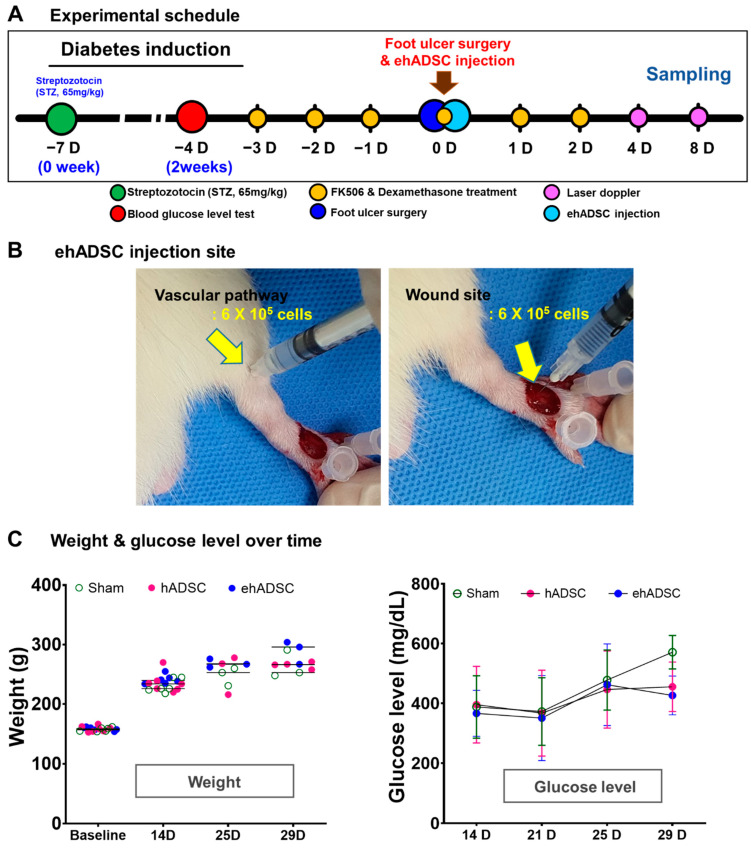
Experimental schedule, body weight, and blood glucose level. (**A**) In vivo experimental schedule. Diabetes was induced by intraperitoneal injection of streptozotocin (STZ, 65 mg/kg) in 6-week-old male Sprague–Dawley rats. On the fourth day of immunosuppressant administration, ehADSCs and hADSCs were injected topically into the foot near the artificial wound. Sampling was performed on days 4 and 8 after administration. (**B**) ADSCs injection site and route. ADSCs were injected around the vascular pathway (left) and near the wound (right). (**C**) Body weight and blood glucose level. There was no significant difference in body weight. There was no difference among groups in blood glucose levels until day 25, and blood glucose levels in ehADSC decreased on day 29, but there was no significant difference among groups.

**Figure 4 cells-12-01146-f004:**
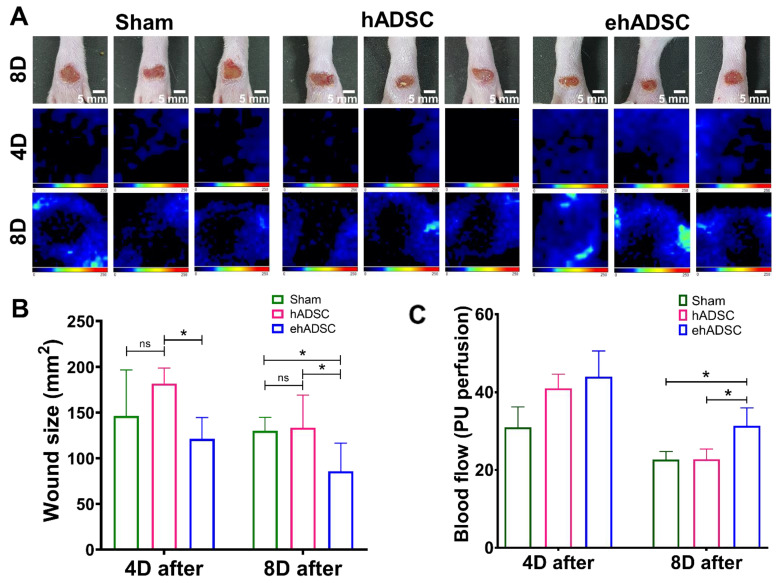
Wound closure and blood flow. (**A**) Wound size and Doppler evaluations were performed at 4 and 8 days after wound formation ADSCs injection. (**B**) At 4 days, a significant difference in wound size was observed between the hADSC and ehADSC groups. At 8 days, wounds in the ehADSC group were still significantly different (smaller) compared to others. (**C**) Blood flow 4 and 8 days after the ADSC injection are shown. There was no significant difference in the blood flow among the groups on day 4. On day 8, the ehADSC group exhibited a significantly increased blood flow compared to others. (* *p* < 0.05; ns, not significant).

**Figure 5 cells-12-01146-f005:**
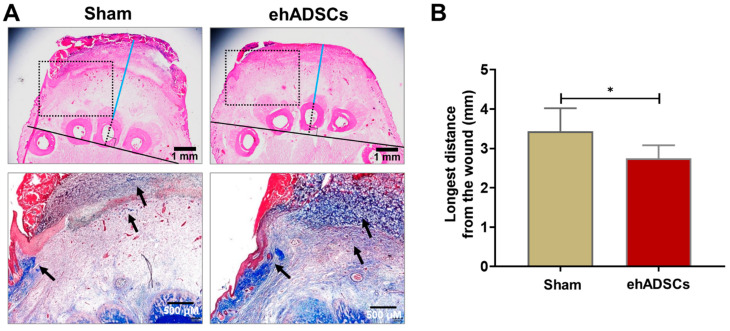
Wound histology. (**A**) Histological evaluation of wounds by H&E (soft tissue thickness between the epidermis and bone is a line drawn horizontally and vertically from the lower end of the animal’s 3rd toe bone and marked in blue from the upper end of the bone to the lower part of the wound) and Masson’s trichrome (dotted rectangle; collagen deposition was stained blue: black arrow) staining at 8 days. A tightly connected tissue structure and greater collagen deposition were observed in the ehADSC group, compared with the sham group. (**B**) The ehADSC group tended to exhibit a shorter bone-epidermis length, compared with the sham group (* *p* < 0.05).

**Figure 6 cells-12-01146-f006:**
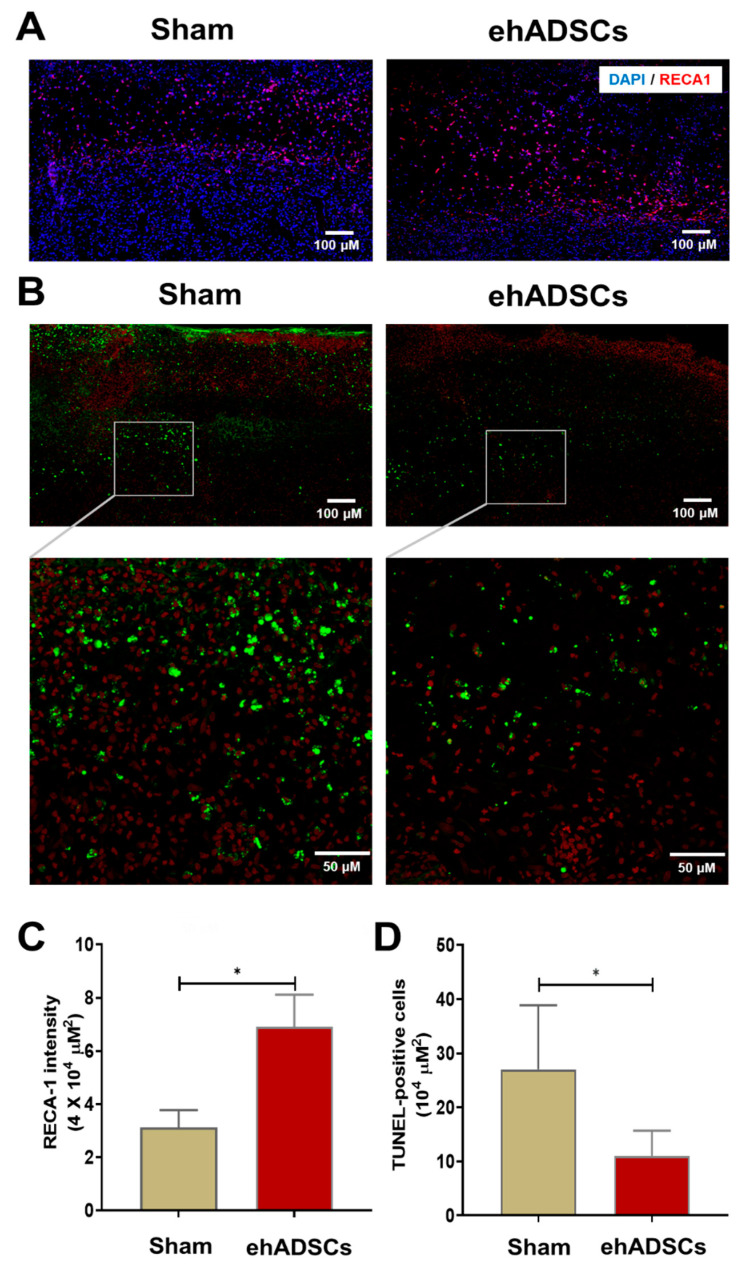
Epifluorescence analysis and TUNEL assay. (**A**) The ehADSC group exhibited greater RECA-1 staining intensity than the sham group on day 8. (**B**) The ehADSC group exhibited fewer TUNEL-positive cells than the sham group on day 8. (**C**) RECA-1 staining intensity in the wound area was significantly greater in the ehADSC group than in the sham group. (**D**) The number of TUNEL-positive cells in deep soft tissue was significantly lower in the ehADSC group than in the sham group (* *p* < 0.05).

**Figure 7 cells-12-01146-f007:**
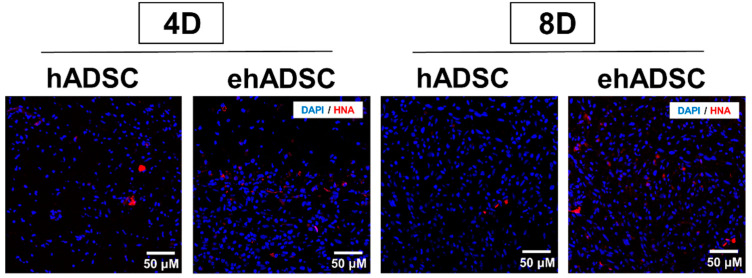
Observation of HNA-positive cells. Representative images of HNA-positive cells are shown at both 4 and 8 days after transplantation.

**Table 1 cells-12-01146-t001:** Summary of outcomes in each animals with ADSC transplantation.

Group	On Day	Animal No.	Wound Size (mm^2^)	Blood Flow (CPU)	HNA Expression	
hADSC	4 D	#10	196.2	37	Negative	1 out of 3 positive
		#11	186.2	44	Positive	
		#12	163.4	42	Negative	
	8 D	#1	145.5	21	Negative	1 out of 4 positive
		#2	86.5	25	Negative	
		#3	130.9	25	Positive	
		#4	171.4	20	Negative	
ehADSC	4 D	#1	160.2	50	Negative	2 out of 3 positive
		#2	115.5	45	Positive	
		#3	109.0	37	Positive	
	8 D	#4	65.9	34	Negative	2 out of 3 positive
		#7	86.1	34	Positive	
		#9	64.5	26	Positive	

## Data Availability

The authors confirm that the data supporting the findings of this study are available within the article. Raw data that support the findings of this study are available from the corresponding author upon reasonable request.

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
