# Peer review of "Human Fibroblast Growth Factor-Treated Adipose-Derived Stem Cells Facilitate Wound Healing and Revascularization in Rats with Streptozotocin-Induced Diabetes Mellitus"

_cells, 2023, doi:10.3390/cells12081146_

Round 1
Reviewer 1 Report
The manuscript does not deliver novel biological data and is not suitable for Cells publication.
Authors describe the impact of FGF-primed (enhanced") human ADSC on wound healing in diabetic rats. Since the effects of mesenchymal stromal cells on diabetic wounds were previously published, the novelty of the proposed paper would be the enhancement of hADSC activity resulting from their priming (in vitro culture) witch FGF. Unfortunately, the authors compared hADSC and ehADSC only in vitro and only in one timepoint, directly after ehADSC enhancement. The crucial missing data are the comparison of hADSC and ehADSC in vivo effects on diabetic wound healing, and missing data on the presence of ADSCs in wound region after their delivery during the time being of the observation.
Detailed remarks
- how long lasted the "enhancement effect" of ADSC?
- Do the "passage 2 cells" (line 99) is the same as "3rd subculture (Fig. 1.).
- line 150 (migration assay). The scratch test is commonly accepted for the migration control purpose in nonproliferating or slowly proliferating cells. ADSCs, however, have strong proliferation potential at the stage of established 2D culture, and the distinguishment between proliferation of migration by scratch reduction would need alternatively marking of cells at the stage of scratching, od monitoring the process by time-lapse photography.
- Why there is no control group receiving "non-enhanced" ADSC from the same donor? Without such control, any speculation on the effects of "enhancement" on ADSC in vivo effects is worthless. The experiment, performed by the other group (Seo et al. 2017) on another species (mice, not rat) can not substitute the real control group missing in the experiment's design.
- Authors do not present any data confirming the prolonged presence of living ehADSC in wound site, they did not try to label, track these cells or find them in immunohistochemical sections from wound site.
- the comparison of hADSC and ehADSC should cover the period equivalent to the duration of in vivo experiment. Presented data do not document, how long after cessation of the exposure of cultured hADSC to cytokines lasts the "enhancement" effect?
Reviewer 2 Report
The study investigated the therapeutic potential of HFGF-enhanced ADSC in wound recovery in diabetes.
My comments are as follows:
1. To indicate Research Question and gap of study in the Introduction. Why is there a need to incorporate HFGF into ADSC?
2. Recent works have not been cited:
Sun Y, Song L, Zhang Y, Wang H, Dong X. Adipose stem cells from type 2 diabetic mice exhibit therapeutic potential in wound healing. Stem Cell Res Ther. 2020 Jul 17;11(1):298. doi: 10.1186/s13287-020-01817-1. Erratum in: Stem Cell Res Ther. 2022 Feb 25;13(1):82. PMID: 32680569; PMCID: PMC7368682.
Stachura, A.; Khanna, I.; Krysiak, P.; Paskal, W.; Włodarski, P. Wound Healing Impairment in Type 2 Diabetes Model of Leptin-Deficient Mice—A Mechanistic Systematic Review. Int. J. Mol. Sci. 2022, 23, 8621. https://doi.org/10.3390/ijms23158621
3. To provide ethical clearance reference no in the Methodology section.
4. Exclusion criteria for patient selection shall be clearly written
5. Typo ? Pprimary culture of ehADSCs
6. To provide references for cell culture and bioassays / established protocols in the Methodology
7. To include brand name, state and country's name for equipments and consumables in the Methodology
8. Number of cells, condition and duration of incubation, choice of multiwell plate should be clearly written in the protocol
9. How was 65mg/kg STZ selected?
10. To describe the methods for animal leg amputation and choice of anaesthetic agent
11. What is sham group? Animal groupings must be clearly stated in the Methodology
12. What is the dose of ehADSCs used in aminal study? How was the dose selected?
13. State the reason for not including non-enhanced HDASC and positive control in the study.
14. To label histological structures in Figure 5A, indicate soft tissue and collagen deposition. Expand the caption. State the source of tissue/organ for Wound Histology
15. State the definition of soft tissue. Does H&E but not Masson Trichrome stain the soft tissue?
16. Table 1 should not be placed in the Discussion. Please discuss Table 1 in the text.
17. There are limitations in this study – selection of ehADSCs dose, administration route of STZ, choice of rodent models (eg C57BL/6J mice fed with a high-fat diet), translational research involving humans, unfavourable animal condition throughout the study.
18. Missing conclusion
Reviewer 3 Report
The topic is interesting, especially wound healing development in diabetes or immune-deficient patient. However, this manuscript has to undergo major revision prior to acceptance.
1) 70% of the abstract contains real facts about diabetes mellitus and cell therapies, and the results from the results are very little. The abstract has to be rewritten with a focus on quantitative results.
2) The authors have characterized ADSCs by flow cytometry. A) where is the FACS data? It should be available as supplementary. B) It would be good if the authors confirm the FACS data with at least one genetic marker in real-time PCR.
3) For the migration test, it would be nice to scan the two groups of the cells at 0h and 10h by AFM. This provides the mechanical properties difference between the two groups.
4) Why sterility test has been carried out based on culturing approach? DNA content would be more accurate, since some mycoplasma could contaminate mammalian cell lines, however, cannot be cultured easily.
5) Authors should justify/explain the difference between hADSCs and ehADSCs (Fig 2A)
6) Why the authors used STZ at 65mg/Kg? is there any optimization? I suggest using or comparing their method with one of the previous articles as ref. https://link.springer.com/article/10.1007/BF02913315
7) Why the error bars in fig 4, especially fig 5C is very high?
8) Authors claimed ehADSCs reduce apoptotic debris. There any cell/molecule marker been used?
Round 2
Reviewer 1 Report
I find the improvements included by the authors satisfactory, the manuscript is suitable for publication.
Reviewer 3 Report
The authors performed extensive corrections on their MS and the quality of the work has been increased and would recommend publishing.